# The 3D-McMap Guidelines: Three-Dimensional Multicomposite Microsphere Adaptive Printing

**DOI:** 10.3390/biomimetics9020094

**Published:** 2024-02-06

**Authors:** Roland M. Klar, James Cox, Naren Raja, Stefan Lohfeld

**Affiliations:** Department of Oral and Craniofacial Sciences, School of Dentistry, University of Missouri-Kansas City, Kansas City, MO 64108, USA; jccvmc@umsystem.edu (J.C.); nrw8q@umkc.edu (N.R.)

**Keywords:** 3D bioprinting, microspheres, multicomposite scaffold, PLGA, PLA, Bioplotter

## Abstract

Microspheres, synthesized from diverse natural or synthetic polymers, are readily utilized in biomedical tissue engineering to improve the healing of various tissues. Their ability to encapsulate growth factors, therapeutics, and natural biomolecules, which can aid tissue regeneration, makes microspheres invaluable for future clinical therapies. While microsphere-supplemented scaffolds have been investigated, a pure microsphere scaffold with an optimized architecture has been challenging to create via 3D printing methods due to issues that prevent consistent deposition of microsphere-based materials and their ability to maintain the shape of the 3D-printed structure. Utilizing the extrusion printing process, we established a methodology that not only allows the creation of large microsphere scaffolds but also multicomposite matrices into which cells, growth factors, and therapeutics encapsulated in microspheres can be directly deposited during the printing process. Our 3D-McMap method provides some critical guidelines for issues with scaffold shape fidelity during and after printing. Carefully timed breaks, minuscule drying steps, and adjustments to extrusion parameters generated an evenly layered large microsphere scaffold that retained its internal architecture. Such scaffolds are superior to other microsphere-containing scaffolds, as they can release biomolecules in a highly controlled spatiotemporal manner. This capability permits us to study cell responses to the delivered signals to develop scaffolds that precisely modulate new tissue formation.

## 1. Introduction

From the 1970s, microspheres and their potential uses for drug delivery have been thoroughly investigated [1,2]. However, their direct use in scaffold design or supplementation of synthetic or natural biomaterial scaffolds with medications or growth factors to boost healing as part of the regenerative medical tissue-engineering field has only been recognized within the last two decades [3,4]. The fabrication of microsphere-based scaffolds can be achieved over various assembly and sintering techniques, each of which can affect the in vivo or in vitro properties of the microspheres and the scaffold they are a part of [5,6]. Compared to conventional tissue-engineering scaffolds, microsphere-based scaffolds exhibit numerous advantages. The major advantages are that microspheres provide control over spatial and temporal release of bioactive factors, which can provide unique cues to stem cells for their differentiation to form the desired tissues, from a structure with an inherent porosity, which benefits vascularization and fluid flow through the structure [7,8,9,10].

However, current techniques cannot distribute microspheres in a desirable fashion because these spheres are unable to be printed directly on their own and hence do not allow for targeted deposition within a 3D structure. When microspheres are used in scaffolds, they are generally entirely buried in another phase, losing the collective advantage of their shapes, properties, and high surface-to-volume ratios, and are available in rather small quantities compared to the volume of the scaffold [11,12,13]. In such instances, they also require the embedding material to degrade first before they can act. Furthermore, the extrusion processes often used for printing scaffolds typically cannot create within the printed struts a microporosity beneficial for cell attachment and proliferation [14,15,16]. A 3D-printed scaffold consisting solely of microspheres would provide the highest control over releasing substances and, with specific intrinsic geometrical configurations, support cyto-differentiation. Having such a scaffold at hand would eventually lead to better tissue-engineering therapies.

Additionally, multicomposite microsphere-based scaffolds for an optimized regeneration of complex tissue structures, e.g., the osteochondral structure of a condyle with both an articular cartilaginous and subchondral osteogenic layer, are elusive [8]. The main difficulty to directly print purely microsphere-based scaffolds has to do with the relation of particle size to the nozzle diameter, which impacts the flow characteristics. To counteract this issue, microspheres are typically mixed with low viscosity materials for 3D printing, e.g., gelatin-like gels [17]. However, for larger scaffolds with multiple layers, the increasing weight of the added layers onto the first few ones can lead to a deformation of the scaffold structure when the viscosity of the material is too low [18]. On the other hand, choosing a higher viscosity carrier may cause higher friction within the ink, leading to blockage of the solid particles and primarily extrusion of the liquid phase, which, in turn, reduces the flow capability of the entire ink over time. If these limitations can be overcome, superior 3D-printed microsphere-based scaffolds with cyto-bioactive properties would be more readily available. 

The aim of the present study was thus to develop an optimized 3D bioprinting method/guide utilizing microspheres to consistently fabricate multicomposite scaffolds. Additionally, the printed scaffolds should possess intrinsic geometric structures that would support cell attachment, migration, and differentiation, thereby fully exploiting the capabilities of microspheres for tissue-engineering applications.

## 2. Materials and Methods

### 2.1. PLA Microsphere Production (200 µm)

Microspheres with an average diameter of 200 µm were produced from poly(lactic acid) (PLA; MIKA3D Filament store on www.amazon.com (accessed on 9 April 2023)) filaments and poly(lactic-co-glycolic acid) (PLGA; Corbion, Amsterdam, The Netherlands) crystals, respectively, using a Büchi Encapsulator B-390 (Büchi Labortechnik, Flawil, Switzerland). Briefly, an inner core nozzle with 200 µm diameter and an outer shell nozzle with 300 µm diameter were mounted on the microsphere manufacturing unit of the Encapsulator. The core fluid consisted of 5% (*w*/*v*) PLA (MIKA3D) or PLGA (Corbion) dissolved in dichloromethane (Thermo Fisher Scientific, Waltham, MA, USA). The shell fluid consisted of 0.33% (*v*/*v*) poly(vinyl alcohol) (Polyscience Inc., Warrington, PA, USA). The nozzle assembly was immersed in a 0.33% polyvinyl alcohol solution (Figure 1A) that was agitated using an Isotemp stirrer platform (Thermo Fisher Scientific) set at 200 rpm to prevent generated microspheres from lumping. Flow rates through the nozzle assembly were set to 4 mL/min (shell fluid) and 2 mL/min (core fluid), respectively, using syringe pumps (KD Scientific Inc., Holliston, MA, USA). The vibration unit was set to a frequency of 1000 Hz and an amplitude value of 6. These parameters were found following the guidelines for the Büchi Encapsulator to generate microspheres in the desired size range between 180 µm to 220 µm for the used polymer solutions.

After microsphere production, the collection solution was stirred for another 8 h to allow for dichloromethane evaporation and microsphere hardening. The microspheres were then thoroughly washed in distilled water, extracted, and lyophilized in a FreeZone 4.5Plus lyophilizer (Labconco Corporation, Kansas City, MO, USA). The dried microspheres were stored at −20 °C until further use.

### 2.2. Microsphere Quality Control

For better control of release from the microspheres in tissue-engineering applications at a later stage, we aimed to print with monodisperse microspheres with a diameter of 200 ± 15 µm. Hence, prior to being utilized in the 3D-bioprinting process, the microsphere batches were sifted to ensure that at least 98% of the used microspheres fell into this size range. USA standard sieves (Anylia Scientific, Vernon Hills, IL, USA) with mesh sizes of 212 µm and 190 µm, respectively, were utilized to separate the desired microsphere size range. As PLGA microspheres possess high electrostatic forces [19] and tend to stick to the sieve’s wall, sucrose (Thermo Fisher Scientific) was milled into particles (~40 µm) and added at a ratio of 1:1 to the microspheres. Sifting was performed under agitation with a No. 1A Vibrator (Buffalo Dental Inc. Syosset, NY, USA). Sifted microbeads were then washed in distilled water to eliminate sucrose and analyzed under a fluorescence microscope (Keyence BZ-X800, Keyence Corporation, Osaka, Japan) to validate quality and sucrose removal. After the removal of the sucrose was confirmed by use of fluorescence microscopy, the washed microspheres were extracted, lyophilized, and then stored at −20 °C until bioprinting.

### 2.3. Bioink Preparation

A solution of three percent carboxymethyl cellulose (CMC, Sigma-Aldrich, St. Louis, MO, USA) in DI water was prepared and mixed with the microspheres to produce the bioink. The ratio of PLA microspheres to 3% CMC was 5:4 (*w*/*w*), whereas the ratio was 5:3 (*w*/*w*) for PLGA microspheres. These ratios were found to be the best for use when extruding 200 µm +/−10 µm PLA or PLGA microspheres through 16G and 18G, respectively, syringe tips, as used during 3D printing. The bioink was loaded into a 1 mL Luer lock syringe (Thermo Fisher Scientific) utilizing a custom rig (Figure 1(B,C1,C2)) to fit into the standard 30 mL syringes normally utilized in the 3D-Bioplotter system.

### 2.4. Three-Dimensional (3D) Printing

For 3D printing, the extrusion-printing method [20] was employed with an EnvisionTec 3D-Bioplotter system (EnvisionTec, Gladbeck, Germany). 

To produce PLA specimens, an 8 mm × 10 mm cylinder (diameter × height) single-color PLA microsphere scaffold and an 8 mm × 10 mm (diameter × height) hemisphere scaffold consisting of differently colored PLA microspheres were manufactured. The three-dimensional structures were first designed in Perfactory Suite v3.1 (EnvisionTec) and then imported into the Visual Machines v2.1 software (EnvisionTec) of the 3D-Bioplotter. Following the recommendations of the printer’s manufacturer, layer thickness was set to approximately 80% of the inner diameter of the syringe tip used for printing. In this case, by using an 18G tip, the layer thickness for this model was set to 0.67 mm. For both the cylinder and the hemisphere, a fill pattern consisted of continuous lines with 2.0 mm distance between their centerlines. The contour was printed with a single line. Each layer was rotated by 90° to the previous one. The PLA microsphere-based bioinks were extruded through an 18 Gauge (18G; (Nordson EFD, Nordson Corporation, Westlake, OH, USA)) precision syringe tip.

For the PLGA specimens, a cylinder of 5 mm × 2 mm (diameter × height) was printed using PLGA-microsphere-based bioink. The shape was designed and printed as described above for PLA; however, the PLGA microsphere-based bioink was extruded using a 16G precision syringe tip (Nordson EFD), and the layer thickness was set at 0.95 mm.

Utilizing the built-in “Material parameters Tuning/Optimization” tool of the Visual Machines software, the starting printing parameters for both PLA and PLGA were set as follows: extrusion printing was performed with a low-temperature print head set to 21 °C; needle offset was set at 0.95 mm (for the 16G tip) and 0.67 mm (18G); starting extrusion pressure was 3.0 bar; printing speed was 1.5 mm/s; and the printing stage temperature was left at room temperature. PLGA and PLA scaffolds were printed on polyimide tape (Tapes Masters store on www.amazon.com (accessed on 9 April 2023)) and left to dry for 24 h before removal. 

### 2.5. Vapor Sintering (Dichloromethane) 

Once dried, 3D-printed PLGA scaffolds were sintered in dichloromethane vapor. Scaffolds were placed into a custom-designed vapor sintering chamber (Figure 1D) that allowed for proper dichloromethane gas penetration. PLGA scaffolds were sintered for exactly 165 s +/−0.5 s, whereas PLA scaffolds required a sintering time of exactly 1200 s +/−0.5 s. Sintering effects were analyzed scanning electron microscopy in conjunction with micro-computed tomography and mechanical testing. 

### 2.6. Scanning Electron Microscopy

Secondary electron (SE) images were acquired at high vacuum mode using a Philips XL30 ESEM-FEG environmental scanning electron microscope (SEMTech Solutions, North Billerica, MA, USA). The samples were coated with Au-Pd alloy for 60 s. The experimental conditions were an accelerating voltage of 15 kV, spot size 4, and a working distance of 20 mm. The digital images were used for image analysis.

### 2.7. Micro-Computed Tomography

PLA or PLGA scaffolds were placed in a custom Styrofoam holder and individually imaged at 18 µm isotropic resolution using a Skyscan1275 (Bruker Corporation, Billerica, MA, USA). The following scan settings were used for all imaging: 55 V, 180 μA, 45 ms exposure, 360° imaging, 0.2° rotation step, and six-frame averaging.

The raw images from each scan were then reconstructed using NRecon software (v1.7.4.2; Bruker Corporation) and imported into Drishti volume exploration software (v3.0.0; https://github.com/nci/Drishti (accessed on 22 May 2023)) for 3D rendering. The rendering settings were optimized for the visualization and assessment of microspheres. 

### 2.8. Mechanical Stability Testing

PLGA cylindrical scaffolds were tested for mechanical properties using a uniaxial compression test machine (Instron 5967 Dual Column Universal Testing System, Instron, Norwood, MA, USA). A 30 kN load cell was used with a testing speed of 1 mm/min. Testing samples were prepared using extrudable pastes consisting of PLGA microspheres 200 µm +/−10 µm and 3% CMC (mixing ratio of 5:4 (*w*/*w*). Dried cylindrical samples (4.5 mm diameter and 3 mm height) were either left unsintered or were vapor sintered with dichloromethane. Vapor sintering was performed by exposing the samples to dichloromethane vapor for 165 s +/−0.5 s for “sintered” samples and for 240 s for the “oversintered” samples. These samples (*n* = 3) were compressed to a final height of 1 mm (i.e., 2 mm of compression). The stress–strain curves of sintered and oversintered samples were compared to unsintered samples to see the effect of sintering on the mechanical stability of the cylindrical samples. 

## 3. Results

### 3.1. Generation of 200 µm Diameter Microspheres

Analysis of produced microspheres from PLA filaments or PLGA crystalline powder revealed similarities and differences in microsphere properties. Both materials fluoresce green under fluorescence light microscopy (Figure 2A,B). Under the current Encapsulator production settings, the average yield of microspheres of a diameter of 200 µm +/−10 µm was 75%, with each PLA or PLGA batch producing, on average, 1 g of microspheres. To remove the remaining 25% of unwanted microsphere “contaminants”, the spheres were sifted using custom-sized sieves to receive a proper monodispersion. During sifting, it was observed that PLA microspheres did not show notable electrostatic charges, allowing for easy separation (Figure 2A). The opposite was observed for PLGA-derived microspheres (Figure 2B). To achieve proper separation of contaminating PLGA sphere sizes, sucrose was used as a sifting agent to overcome electrostatic forces [21]. Fluorescent microscopy confirmed sucrose presence, as the autofluorescence of the PLGA was blocked (Figure 2C, white rings mark the microspheres). The PLGA microspheres were coated by the sucrose powder, which enabled proper sifting and separation of unwanted spheres not of the size 200 µm +/−10 µm. After the sifting, 200 µm +/−10 µm PLGA microspheres were washed, and sucrose absence was validated through re-establishment of the PLGA autofluorescence signal (Figure 2D). 

### 3.2. 3D-McMap Method/Guide

The first 3D-bioprinted specimens were manufactured under standard printer settings. These included:A continuous uninterrupted printing cycle;Printing stage temperature remained at 21 °C;Extrusion pressure was unaltered;Needle offsets were not varied. 

Under these standard printing conditions, it was observed that all scaffolds collapsed under their own weights, losing both shape and internal architectural parameters (Figure 3(A1–A3)). 

Several changes that occurred during 3D bioprinting of the microsphere-based bioink under standard conditions were noticed:The pressure required to extrude the ink changed during the printing process, often increasing by 0.5–1 bar per extruded 0.25 mL of bioink. An analysis of the material showed that the microsphere ink was drying out in the syringe, losing its flow characteristics. Altering the specified ratio of microspheres to CMC and/or altering the concentration of the CMC prevented the proper flow of the ink.After two layers, the added weight of subsequent layers onto previous ones caused the first two layers slowly to collapse, as the relative wettish nature of the bioink could not withstand the pressure. During this process, any pores created by design and printed into the scaffolds were filled (Figure 3(A2,A3)).On the non-heated print bed, the printed layers did not dry quickly enough to stabilize their shapes to prevent the issues raised in point 2.Scaffolds printed on polyimide tape for better adhesion during printing stuck strongly to the tape after 30 min drying time already and could not be removed from the platform without breakage.

Based on these initial results, the following subtle changes were implemented through our 3D-McMap protocol: Cooling of the printhead with the microsphere/CMC bioink to 4 °C ensured that the bioink continued to properly lubricate the microspheres, which greatly reduced the need for extrusion pressures adjustments during printing. A small pressure change was required only after two layers had been printed. Thereafter, the extrusion pressure needed no further changes. It was observed that only the bioink at the tip of the extrusion syringe, which was outside the cooled area, dried out over time. This issue was counteracted by pausing the printing process as necessary and briefly dabbing it with a sterile water-wetted tissue paper for 5 s.The print stage was kept between 50 °C and 60 °C, and the printing process was paused every two layers of each shape for up to 10 s. This action resulted in a distinct improvement in the stability of both external and internal geometric structures (Figure 3(B,C1–C3) and Figure 4) despite the added weight of additional layers.The implementation of this drying step, however, also resulted in shrinkage of 0.1–0.15 mm in the printed scaffold every two layers, causing the follow-on layers to be out of alignment. To compensate for this, a cylinder was designed that comprised multiple-cylinder sections, with each section being two layers thick and taking into consideration the 0.1–0.15 mm shrinkage of the drying step. This ensured that all layers connected up properly, and the final 3D structure maintained its overall shape-integrity, producing a symmetrical 3D-printed shape each time (Figure 3(B,C1)).The polyimide tape was replaced with aluminum foil. The printed structure could easily be removed from the foil, even if only partially dried after 30 min. This made the collection of scaffolds very simple.

The µCT image in Figure 4 shows the high density of microspheres in the struts. Gaps between microspheres and missing spheres contribute to the microporosity of the scaffold.

### 3.3. Multicomposite 3D-Printed Hemisphere

Without the implementation of our 3D-McMap protocol, printing complex 3D structures, such as hemispheres, with microsphere-based inks resulted in a collapse of the internal strutted superstructure and consequently of the overall 3D structure (Figure 3(A2,A3)). As with the cylinders composed of a single-colored microsphere type, the softness of the lower layers and the added weight of subsequent layers contributed greatly to the collapse of the hemisphere structure. However, after implementing the 3D-McMap method/guide, there was a significant improvement to both the overall symmetry of the hemisphere scaffold and the internal strutted structure (Figure 5). Hence, when printing with multiple microsphere materials, represented here by the different colors, highly complex multicomposite scaffolds could be printed possessing typical shape and composition characteristics that could mimic in vivo biological structures, specifically those of a joint condyle (articular cartilage with internal subchondral bone). 

### 3.4. Sintering Periods and Mechanical Stability 

To improve the stability of the final microsphere-based scaffold, particularly for load-bearing applications, the scaffolds were exposed to DCM vapor for sintering. PLGA microsphere scaffolds were sintered for 165 s. This sintering duration was previously observed to cause the desired sintering degree as depicted in Figure 6B, whereas longer sintering times cause over-sintering with microspheres melting, merging into a single structure, and significantly reducing the microporosity within the structure (Figure 6C). 

Mechanical tests were conducted using sintered, over-sintered, and unsintered (control) cylindrical samples comprising PLGA microspheres and CMC polymer. A comparison of the stress–strain curves of the tested samples is shown in Figure 7A,B under continuous compressive load. Unsintered samples displayed a brittle behavior and showed a clear yield point at 3.5 MPa, after which the sample broke into pieces (Figure 6D). Contrary to unsintered samples, sintered samples and even oversintered samples showed continuous increases in compressive stress with increasing strain. However, oversintered samples show lower compressive moduli. This was confirmed by comparing the tested samples, as seen in Figure 6E. Sintered samples had compressive strength of 13.6 MPa (Figure 6F), which was more than double the strength of oversintered samples (5.57 MPa) and unsintered samples (3.8 MPa), within the test conditions (i.e., 2 mm compression).

## 4. Discussion

The future of tissue engineering depends strongly on scaffolds that replicate the intrinsic biological structures in conjunction with proper spatiotemporal signaling to promote correct tissue formation. In his review regarding the failure and possible solutions for regenerating bone clinically, Klar [22] described a simplistic approach that tissue engineers have yet to reproduce. It is stated that one needs to reverse engineer tissue types into their separate components, reproduce these components synthetically, and then recombine them to form a functional synthetic–biological replicate that behaves and responds exactly like the in vivo counterpart [22]. The ideal scaffold has good biocompatibility, tunable degradability, and an interconnected porous structure that mimics the architectural and mechanical properties of the tissue and, most importantly, contains the correct chronological signaling setup within its layered macro- and micro-porous architecture to guide proper tissue formation [22,23,24]. Properly defining the correct signaling impetus is one of the most challenging tasks in tissue engineering and regenerative medicine. Highly controlled microsphere scaffolds have great potential to solve the issue with determining the optimal signaling cascade towards proper tissue formation [25,26,27].

Microspheres have so far successfully been used when supplemented into scaffolds. In recent years, an increasing number of studies have emerged to develop bioactive systems by combining microspheres with bioactive molecules, which have been demonstrated to not only provide bionic biodegradable physical support for tissue growth but also allow for the secretion of biological molecules to regulate tissue reformation [28,29]. For example, hydroxyapatite microspheres have been successfully cross-linked with collagen matrices and/or chitosan to fabricate a composite scaffold that has both in vitro and in vivo capabilities on the regeneration of bone, allowing for better spatial mechanical stability and enhances the regeneration of bone in tissue defects [25,28,30]. Similar studies for cartilage regeneration have seen the emergence of scaffolds that are both flexible and soft enough, composed of simple biphasic calcium phosphate granules, hyaluronic acid-gelatin hydrogel, and polydopamine PLGA microspheres, which are able to increase the overall mechanical strength of the composite scaffold, replicating almost natural articular cartilage [31,32,33].

While growth factors have also been incorporated into microspheres that have then been added to composite scaffolds, such as PLGA microspheres loaded with kartogenin on the surface layer of scaffold and/or polylysine-heparin sodium nanoparticles loaded with transforming growth factor (TGF)-β_1_ in the transitioning layer [23,34], the proper position in terms of a spatial pattern that systematically and chronologically spurns differentiation, proliferation, maturation. and transformation of cellular entities to form the correct tissue types remains elusive.

The placement of microspheres within scaffolds is still random, possessing few proper chronological gates to modulate and direct proper tissue formation. As stated previously, this shortcoming is partially based on the fact that we still do not know which combinations are necessary for which layers of a scaffold containing microspheres to properly coordinate the cascade of events that would typically lead up to proper tissue or organ development, as seen embryonically [8]. Embryogenesis teaches us that all organs are formed on specific scaffolding events that rely on concentration gradients established by specific genes, proteins, or cellular aggregations [35]. This patterning helps guide true tissue formation and is tightly regulated during tissue regeneration, although, to our eyes, it appears unorganized. However, this configuration is complicated to reproduce, as, within each so-called layer, cells, proteins, and other bioactive entities are positioned in such a way that drives proper tissue formation. Whilst some newer techniques, especially next-generation 4D printing, are in the process of circumventing some of the more macroscopic tissue-related aspects [36,37], reestablishing motor functions through the use of autonomous self-learning bio-robots disguised as artificial tissue, 3D printing with microspheres may be vastly superior to other 3D printing techniques, as microspheres can reproduce the delicate microscopic complexity of the embryogenic patterning process since each microsphere can be uniquely manufactured to possess multiple characteristics [8,38] and positioned to replicate true intrinsic signaling tissue formation responses. Whilst, to date, the position of microspheres within a scaffold remains problematic, our method/guidelines provide critical solutions to tissue engineers to help design pure multicomposite microsphere scaffolds that better assist in synthetically replicating the intrinsic patterning of organs, allowing for better tissue reformation and ultimately superior bio-integration into relevant tissue implant sites, clinically. 

3D-Bioprinting systems are engineered to be efficient, reducing the cost and time for the manufacture of a scaffold [39,40]. However, the sensitivity essential for the formation of biological layers cannot be optimized for each 3D-printing system, hence requiring adaptation for new material developments. Misplacement of molecules, especially in biology, can have severe ramifications for downstream processes that can result in cancer formation, tissue abnormalities, and, in the worst cases, death [41,42]. This flaw in sensitivity during the 3D-printing process was made very evident during our optimization procedure, in which specific limitations made the 3D-printing procedure of a pure microsphere scaffold almost impossible. 

Three-dimensional printing still requires certain adaptations to streamline the printing process to make printing of microspheres a global possibility. Especially, adjustments in the areas of bioink behavioral changes during 3D printing over time during extrusion and within layers were critical to develop the correct solutions in our printing process. The microspheres to CMC carrier matrix ratios used in the experiments, which were PLA:CMC 5:4 (*w*/*w*) and PLGA:CMC 5:3 (*w*/*w*), were critical to achieving suitable flow. However, with time, the bioink dried out, which partially could be linked to the environment, as was previously shown by Gungor-Ozkerim et al. [43] and Schwab et al. [44]. Minor increases in CMC concentration did not solve the extrusion issue under standard RT conditions. Instead, this caused a significant increase in viscosity, making it difficult to extrude the bioink. We discovered that keeping the ink at 4 °C stabilized the flow behavior for a prolonged time. Further drying of the ink inside the syringe tip was caused by keeping the tip near the heated printing stage during waiting periods, which were added into the printing method/guide to allow the extruded layers to dry. Hence, the intention of heating the printing stage to 50–60 °C to quickly dry and thereby stabilize the shape of the extruded microsphere ink was counteracting the extrusion process, as the radiation heat from the stage caused blockage of the tip by drying the bioink close to the orifice. A manual and timed interruption of the print cycle for 10 s after each layer using the “stop cycle” button on the user interface of the printer control software moved the print head away from the stage and out of the heated zone, preventing the dry-out at the syringe tip. This first printing adaptation step to process the investigated microsphere-rich bioinks prevented drying of the ink during the process and cleared the view to further limitations of the printing system.

Certain types of bioinks extrude better under certain pressure conditions [45,46]. However, extrusion pressure depends on factors such as the diameter of the syringe used for the printing, semi-solid vs. low-viscous materials, and the volume of the printed substance [45,47]. It is known that the greater the volume of a liquid is within a tube, the greater the friction/resistance will be on the inner surface of the tube [45,48]. Hence, as the volume decreases the friction will decrease as well, reducing pressure required to push that liquid forward. Translating these criteria into the 3D-printing process meant that the extrusion pressure applied, due to the reducing volume of the microsphere containing bioink, would eventually cause improper scaffold layering thickness, which is known to negatively impact new tissue re-formation in follow-up in vivo/vitro studies [49]. To rectify this drawback in pressure-related changes due to volume loss of print material, we closely monitored the flow of the extruded material during each layer and adjusted the extrusion pressure as necessary during our printing process, ensuring a consistent microsphere flow rate and maintaining an almost uniform extrusion layer thickness across the scaffolds. 

In addition to uniform material extrusion relevant for 3D printing, the mechanical properties of scaffolds printed from the prepared microsphere-based ink were tested. Vapor sintering was used to fuse the microspheres to increase stiffness for load-bearing applications. As seen in the mechanical characterization of samples comprising microspheres sintered at different time periods, unsintered samples cracked with a defined ultimate stress point at 22% strain given the lack of fusion between microspheres. Sintered and oversintered scaffolds did not show a defined ultimate stress point within the testing range in this work. With higher compression (>2 mm or >70% strain), we expect that sintered samples would show a stress–strain curve with an ultimate stress point like that of unsintered samples. However, as we can see in Figure 6E, the sample was already cracked but had maintained shape due to fused microspheres. Yet, the same cannot be said about the oversintered samples, as they may continue to show plastic behavior due to overly sintered microspheres. The lower performance of the oversintered sample may be due to material degradation due to the long exposure to solvent vapor.

With our present 3D-McMap method/guidelines, we successfully took measures to address issues arising over time when printing with microsphere-rich bioinks. However, monitoring the print quality and adapting the parameters accordingly required two users to assess and manually adjust the process on-the-fly. Eventually, machine learning will need to be incorporated into the hardware that can adapt the printing process automatically on the go, in which 3D-printing machines learn to be gentler during the printing process. The lack of printing gentleness was a very apparent limitation during the printing of the continuous lines within the present studies scaffolds that formed the internal struts. As previously noted by Lee et al. [45] when utilizing other bioinks, the present study also showed that the extruded material was often still very sticky. Whilst this is not necessarily detrimental during the printing of the layers, as it would foster better attachment of the layers to each other and allow for better distribution of microspheres within the layers, it was problematic at the end of each printed layer. There was no proper cut-off at the end of a printed line from the precision syringe tip. This also made printing of plain non-continuously extruded lines impossible. Once the 3D-Bioplotter has reached the end of the layer, it will quickly and efficiently but also very harshly lift the print head upward to transition to the next layer to be printed. This yanking process coupled with the stickiness of the bioink meant that, especially near the ends of printed lines, gaps occurred that would disrupt the next printed layers correct position in that area. Additionally, it is known that such yanking forces can rupture cells [50], which would be problematic for follow-up studies where cells will be included in the bioink. To circumvent this issue in future studies, we opted for printing a continuous line during our microsphere extrusion process. However, by doing so, the lines at the edge of the layered scaffolds were often thicker than initially programmed. Having thicker edges in certain regions of the designed scaffold may not necessarily be harmful for tissue regeneration of specific organs or damaged biological sites, but it may pose a limitation to accurately position specific microspheres within a scaffold later in order to create material gradients that improve tissue regeneration.

## 5. Conclusions

Our 3D-McMap method/guide provides a standardized process with which highly complex and multicomposite scaffolds possessing different gradients of bioactive agents may be manufactured. With the adjustments made during the 3D printing process, we were able to generate not only single-phase microsphere scaffolds but also complex multicomposite scaffolds with varying porosities and architectures to address everchanging requirements for cell migration and development in defects that span several tissues. The suggested process addressed critical issues with 3D printing of microsphere-rich bioinks. More importantly, by overcoming current limitations in printing such specialized inks, the process paves the way toward highly complex constructs for tissue engineering and regenerative medicine that could allow for better modulation of tissue development through multiple gradients and the localized and controlled release of biomolecules.

## Figures and Tables

**Figure 1 biomimetics-09-00094-f001:**
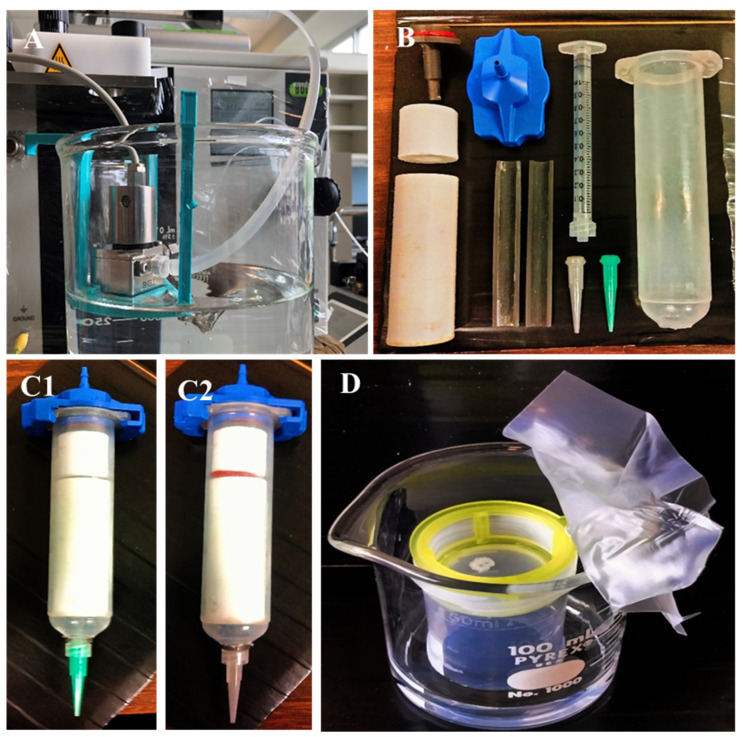
(**A**) Microsphere manufacturing unit with custom carrier platform; (**B**) Custom setup to permit for bio-ink extrusion from a 1 mL Luer lock syringe in a 3D Bioplotter using either a (**C1**) 18G or (**C2**) 16G precision syringe tip; (**D**) Custom dichloromethane vapor sintering chamber.

**Figure 2 biomimetics-09-00094-f002:**
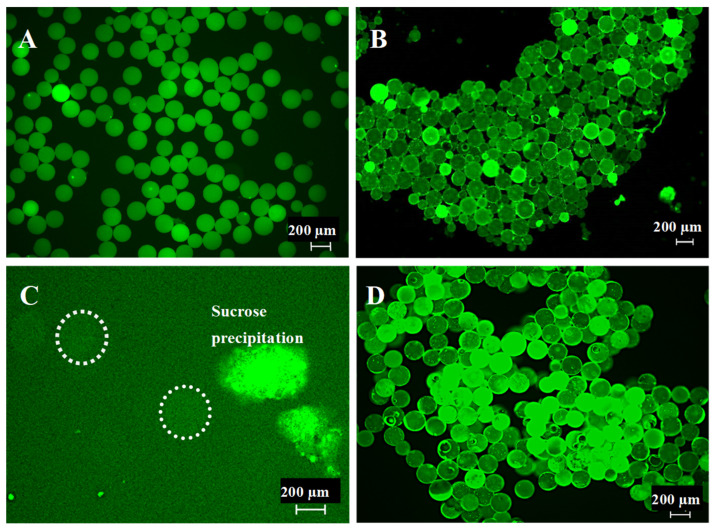
(**A**) PLA microspheres of 200 µm +/−10 µm diameter after sifting. (**B**) PLGA microspheres of 200 µm +/−10 µm diameter after sifting without sucrose. (**C**) Sucrose interference of the autofluoresces capability of PLGA microspheres (dotted circles mark the microspheres). (**D**) PLGA microspheres of 200 µm +/−10 µm diameter sifted with sucrose and after sucrose removal through washing.

**Figure 3 biomimetics-09-00094-f003:**
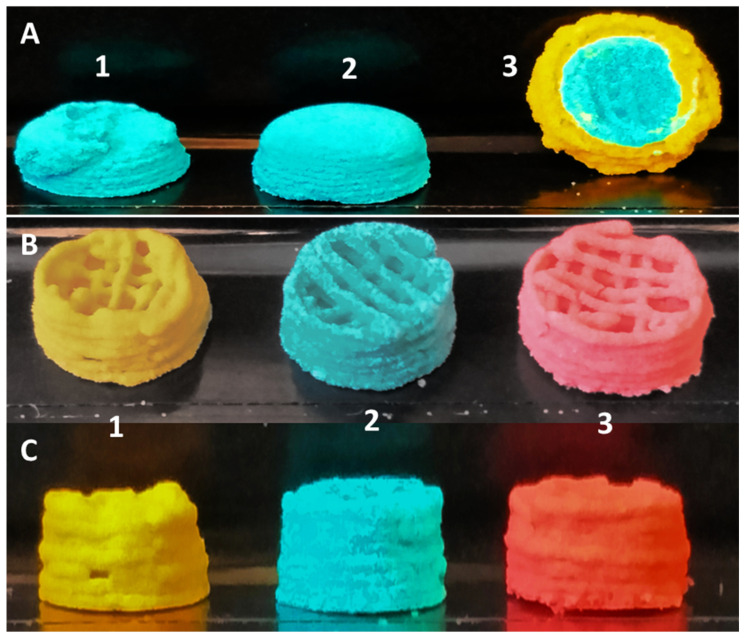
(**A1**–**A3**) Conventional 3D-printing techniques utilizing Ø 200 µm +/−10 µm PLA microspheres versus (**B**,**C1**–**C3**) the 3D-McMap method.

**Figure 4 biomimetics-09-00094-f004:**
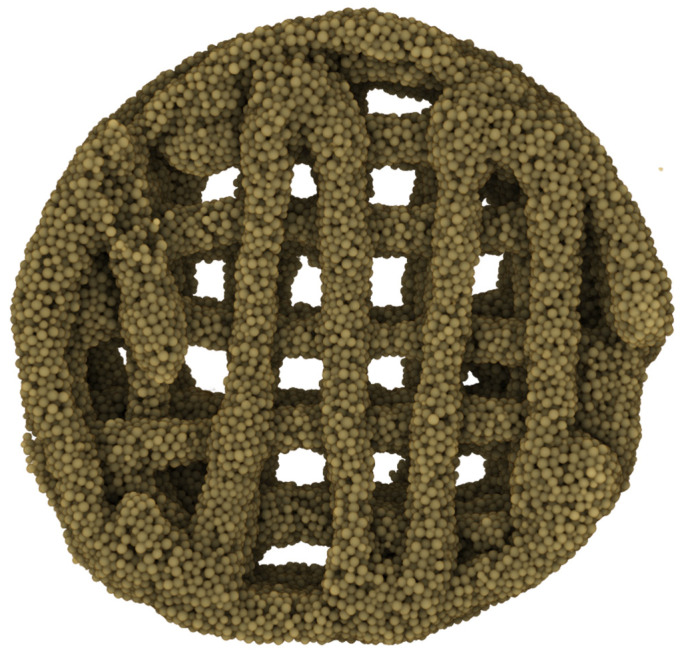
µCT scan (top view) of a PLA microsphere-based scaffold fabricated via the 3D-McMap method.

**Figure 5 biomimetics-09-00094-f005:**
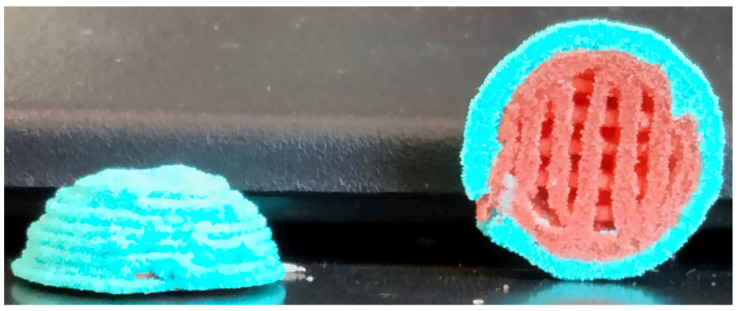
Multicomposite hemisphere/condyle scaffold comprising Ø 200 µm +/−10 µm PLA microspheres, with an internal strutted architecture utilizing the 3D-McMap method/guide.

**Figure 6 biomimetics-09-00094-f006:**
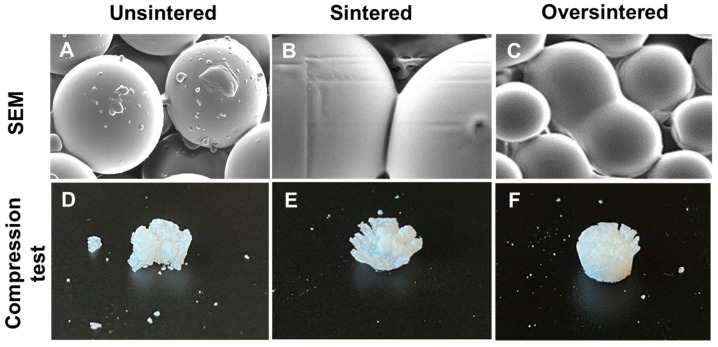
SEM and compression assay structural results of 3D-printed PLGA scaffolds (**A**,**D**) unsintered (control), (**B**,**E**) sintered for 165 s, and (**C**,**F**) oversintered after 240 s.

**Figure 7 biomimetics-09-00094-f007:**
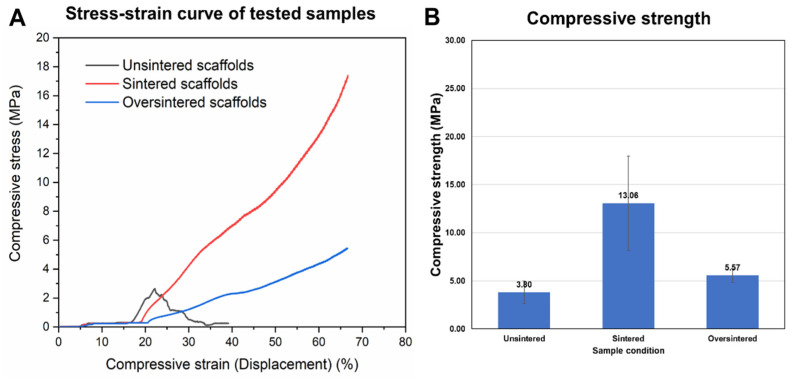
(**A**) Mechanical stability and (**B**) compressive strength of 3D-printed PLGA microsphere scaffolds, unsintered (control), sintered for 165 s, and oversintered for 240 s.

## Data Availability

The data that support the findings of this study are available from the senior corresponding authors upon reasonable request.

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
