# Peer review of "The 3D-McMap Guidelines: Three-Dimensional Multicomposite Microsphere Adaptive Printing"

_biomimetics, 2024, doi:10.3390/biomimetics9020094_

Round 1

Reviewer 1 Report

Comments and Suggestions for Authors

This manuscript “The 3D-McMap protocol: 3-dimensional multicomposite micro-sphere adaptive printing” introduces a way of altering the conventional 3D scaffold printing method to produce a more complex and efficient microsphere containing scaffold. Although this manuscript clearly reiterates and takes on existing problems of conventional 3D scaffold printing, major revision is advised for the following reasons:

1.)   Although there is novelty in the way that 3D scaffold printing used is different from the conventional method, it seems that it only works for PLA and PLGA. More types of polymers should be used to further strengthen the novel method.

2.)   There seems to be too many factors that must be met in order to use this method. Further explanation on why the different factors are important should be added through experimentation for the readers to better understand the principles behind the described method.

3.)   More instances in which this method can be used should be added to strengthen the reason of usage for this method. Currently, it is difficult to know when this method can be used, or if this method is applicable at all.

Author Response

Response to reviewers:

Reviewer 1:

1.)   Although there is novelty in the way that 3D scaffold printing used is different from the conventional method, it seems that it only works for PLA and PLGA. More types of polymers should be used to further strengthen the novel method.

Response: We would like to thank the reviewer for their suggestions. Our proposed guidelines are to help the researcher on printing scaffolds from microspheres of any polymeric composition. While we have used PLA and PLGA for our research, the influence of the material on the actual printing process and its issues is limited and still can be controlled when following the 3D-McMap guidelines. We have tweaked the article and the title slightly to better reflect this.

2.)   There seems to be too many factors that must be met in order to use this method. Further explanation on why the different factors are important should be added through experimentation for the readers to better understand the principles behind the described method.

Response: Thank you for your comment. It is correct that the printing process with microsphere-based inks is depending on many factors. We believe we have appropriately discussed the factors playing a significant role in printing and how issues can be addressed in both the Results section 3.2 and in areas in the discussion (starting from page 11 line 393: “This flaw….”).  We’d like to emphasize that these are guidelines on what to look out for during the printing on other systems when specifically using specific hydrogel carriers.

3.)   More instances in which this method can be used should be added to strengthen the reason of usage for this method. Currently, it is difficult to know when this method can be used, or if this method is applicable at all.

Response: We thank the reviewer for his/her comment. The beginning part of the discussion deals with this part extensively why the use of this method is essential. In a nutshell, it would help us design better scaffolds that can more accurately direct spatial and temporal tissue development. This method is thus applicable for biological and clinical applications regarding regenerative medicine and the design of implants, via tissue engineering, to replicate biological structures. A relevant paragraph on pg. 11 line 375 – 387 has been added to better highlight this before going into the essential factors to look out for and explaining these.

Reviewer 2 Report

Comments and Suggestions for Authors

By employing the extrusion printing technique, this paper developed an approach that enables the creation of large scaffolds made of microspheres. Additionally, this method allows for the direct deposition of cells, growth factors, and therapeutics encapsulated in microspheres into multicomposite matrices during the printing process. The 3D-McMap technology developed specifically tackles the problem of maintaining consistent scaffold shape throughout and after the printing process. Carefully timed breaks, small drying stages, and modifications to extrusion conditions resulted in the creation of a uniformly organized scaffold made of significant microspheres, which maintains its internal structure. This feature allows for the study of cellular responses to the transmitted signals and the development of scaffolds that accurately regulate the production of new tissue. The manuscript is interesting and timely, however, there are issues to be addressed as follows.

1-      Explain how the frequency and magnitude achieved in “The vibration unit was set to a frequency of 1000 Hz and an am-90 plitude value of 6.”

2-      The paper could benefit from a more comprehensive description of the printing parameters (e.g., layer thickness, printing speed, extrusion pressure) and how these parameters were optimized.

3-      The work has modeling and simulation aspects. Include some directions using the following on 3D-printed hydrogels modeling “An anisotropic constitutive model for fiber reinforced salt-sensitive hydrogels” and “3D‐Printed Phase‐Change Artificial Muscles with Autonomous Vibration Control”

4-      Describe the irregular behaviour of Unsintered scaffolds in Figure 6A compared to the sintered and oversintered ones.

5-      Provide some further applications of the proposed methods in cancer and vascularized 4D printing using the works “4D Printing for Vascular Tissue Engineering: Progress and Challenges” and “Recent progress of 4D printing in cancer therapeutics studies”

6-      A thorough analysis of the porosity and internal structure of the printed scaffolds, including how the microsphere arrangement affects these properties, would provide valuable insights into the scaffold's potential for tissue engineering applications.

Comments on the Quality of English Language

Minor editing of English language requiredMinor 

Author Response

Response to reviewers:

Reviewer 2:

1.)   Explain how the frequency and magnitude achieved in “The vibration unit was set to a frequency of 1000 Hz and an amplitude value of 6.”

Response: We would like to thank the reviewer for their question. These parameters were pre-determined during initial optimization tests in which we found that at this frequency and the amplitude in conjunction with the flow speeds of the shell and core fluid syringes, generated a tight microsphere range in which 70-80% of the microspheres came out at a size of between the microsphere size indicated in the manuscript. A small addition has been made on pg. 2 line 91 to better describe this.

2.)     The paper could benefit from a more comprehensive description of the printing parameters (e.g., layer thickness, printing speed, extrusion pressure) and how these parameters were optimized.

Response: We thank the reviewer for pointing out that some of the initial optimization and starting parameters for the 3D printing were missing. We have made additions to section 2.4 3D Printing Parameters that cover the previously missing information. In brief, the 3D bioplotter has a built-in optimization tool that allows for finding optimal printing speed and pressure settings for the selected material. Layer thickness is set to 80% of the inner diameter of the syringe tip used for printing.

3.)     The work has modeling and simulation aspects. Include some directions using the following on 3D-printed hydrogels modeling “An anisotropic constitutive model for fiber reinforced salt-sensitive hydrogels” and “3D‐Printed Phase‐Change Artificial Muscles with Autonomous Vibration Control”

Response: We thank the reviewer for their suggestion regarding the inclusion of some articles. We have included a paragraph covering the two publications suggested by the reviewer in context of this manuscript in the discussion (pg. 11 line 375 – 387).

4.)        Describe the irregular behavior of Unsintered scaffolds in Figure 6A compared to the sintered and oversintered ones.

Response: Thank you for your query. Before answering, we would like to correct the dimensions of the sample mentioned in manuscript. The sample height was 3 mm and diameter was 4.5 mm. By ‘irregular behavior’, we assume you meant that unsintered scaffolds (black curve) have a defined ultimate stress point at ~22% strain compared to sintered (red curve) and oversintered (blue curve) which do not have an ultimate stress point. This can be explained by the sample condition and testing parameters. As mentioned in Pg 5 line 189, we compressed the sample 2 mm i.e. a sample with 3 mm height was compressed to final height of 1 mm. So, the stress strain curve obtained was limited to the testing parameters i.e. 2mm compression (approximately 60-70% strain as seen in the Fig 7A). With higher compression (>2mm or >70% strain), we expect that sintered samples will show a defined ultimate stress point similar to that of unsintered samples. As we can see in Fig 6E, the sample was already cracked but had maintained shape due to fused microspheres. However, the same cannot be said about the oversintered samples as they may continue to show plastic behavior due to over fused microspheres.

We have made the corrections as below:

Section 2.8,

Pg 5 line 191: “…3.5 mm diameter...” to “…4.5 mm diameter…”

Pg 5 line 191: “…4.5 mm height…” to “…3 mm height…”

Pg 5 line 195: “…height of 2.5 mm…” to “…height of 1 mm…”

Included additional details of the testing conditions in results section as below:

Section 3.4,

Pg 9 line 318: “…MPa), within the test conditions (i.e. 2 mm compression).”

Section 4,

Pg 121 line 448: Additional discussion mentioned in the response above has been added.

5.)      Provide some further applications of the proposed methods in cancer and vascularized 4D printing using the works “4D Printing for Vascular Tissue Engineering: Progress and Challenges” and “Recent progress of 4D printing in cancer therapeutics studies”

 Response: We thank the reviewer for their suggestions; however, we believe to be specific on the applications of where our method/guidelines can be used or not is being too speculative. The microsphere-based scaffolds we propose are beneficial for regulating tissue development and are aimed to regenerate more complex tissue types such as bone and articular cartilage, tissue types with inherent complex signaling patterns. The focus of the article, however, is the actual production of 3D printed scaffolds from pure microsphere of any composition. While 4D printed scaffolds are possible with our guideline, this rather depends on the microspheres and their payload. In this context the article is written to globally indicate that this method would generate a microsphere scaffold that generally can heal and regenerate any tissue variant, be it now vascular tissue or possible procedures clinically. Being too specific on applications may mislead the reader, away from the central message which is what to do to generate a pure microsphere scaffold utilizing current 3D printing techniques and what to look out for during the process whilst heavily pronouncing the aspect that we are one of the first groups to generate pure multicomposite microsphere scaffolds.           

6.)      A thorough analysis of the porosity and internal structure of the printed scaffolds, including how the microsphere arrangement affects these properties, would provide valuable insights into the scaffold's potential for tissue engineering applications.

Response: We thank the reviewer for their comment. An additional image and figure legend has been included in the result section to better highlight the internal structure of the printed scaffold (pg 8 Line 277-282). Since the manuscript is providing a guide on how to better print 3D microsphere scaffolds, the discussion of how the current configurations such as porosity and microsphere arrangement would affect tissue engineering applications is beyond the purpose of the manuscript and will be reported in future manuscripts on in vitro and in vivo studies.

Round 2

Reviewer 2 Report

Comments and Suggestions for Authors

The authors have addressed my earlier comments, so the manuscript could be accepted in its current form.

Comments on the Quality of English Language

The authors have addressed my earlier comments, so the manuscript could be accepted in its current form.